# LTD: Low Temperature Distillation for Gradient Masking-free Adversarial Training

## Abstract

Adversarial training has been widely used to enhance the robustness of neural network models against adversarial attacks. Despite the popularity of neural network models, a significant gap exists between the natural and robust accuracy of these models. In this paper, we identify one of the primary reasons for this gap is the common use of one-hot vectors as labels, which hinders the learning process for image recognition. Representing ambiguous images with one-hot vectors is imprecise and may lead the model to suboptimal solutions. To overcome this issue, we propose a novel method called Low Temperature Distillation (LTD) that generates soft labels using the modified knowledge distillation framework. Unlike previous approaches, LTD uses a relatively low temperature in the teacher model and fixed, but different temperatures for the teacher and student models. This modification boosts the model's robustness without encountering the gradient masking problem that has been addressed in defensive distillation. The experimental results demonstrate the effectiveness of the proposed LTD method combined with previous techniques, achieving robust accuracy rates of 58.19%, 31.13%, and 42.08% on CIFAR-10, CIFAR-100, and ImageNet data sets, respectively, without additional unlabeled data.

## 1 Introduction

Deep neural networks (DNNs) have become widely-used tools for challenging tasks such as image classification (Krizhevsky et al., 2012), object detection (Wang et al., 2022), image captioning (Herdade et al., 2019), and semantic analysis (Zhang et al., 2018). The success of these tasks provides the foundation for advanced applications such as self-driving car (Grigorescu et al., 2020), or machine translation (Devlin et al., 2018). However, as DNNs become more prevalent, researchers are exploring practical issues beyond accuracies, such as model compression, unsupervised learning, and robustness.

One critical concern regarding DNNs is their vulnerability to adversarial attacks, which aim to deceive DNNs with high confidence by adding a small perturbation to the input. Adversarial attacks have been observed in several domains, including audio space (Carlini & Wagner, 2018) and text classification (Miyato et al., 2017; Kwon & Lee, 2022) in natural language processing. Adversarial attacks are not limited to the digital world, as they can also occur in the physical world, as shown by cell phone camera attack (Kurakin et al., 2018) or road sign attack (Eykholt et al., 2018; Zolfi et al., 2021). Researchers have also studied backdoor attacks (Yao et al., 2019; Chen et al., 2017b) that reduce model accuracy during training time.

Adversarial attacks can be classified into two types, namely white-box attacks and black-box attacks, based on the amount of information available to the attacker. For white-box attacks, the complete information of target networks is accessible. Examples of white-box attacks include FGSM (Goodfellow et al., 2014), CW attack (Carlini & Wagner, 2017), and AutoAttack (Croce & Hein, 2020). Conversely, for black-box attacks, attackers have access only to the output probability of the target model, and the number of queries is limited. Examples of black-box attacks include Simultaneous Perturbation Stochastic Approximation (SPSA) (Uesato et al., 2018), ZOO (Chen et al., 2017a), and Square Attack (Andriushchenko et al., 2019). Furthermore, adversarial examples generated by a substitute model can fool the target model if the models have similar architectures (Papernot et al., 2017). Generally, adversarial attacks restrict the size of perturbation within

a given $\epsilon$-ball. On the other hand, unrestricted attacks allow for any type of changes to the input data but preserve semantic meaning (Brown et al., 2018).

Adversarial training, one of the most effective defense strategies against adversarial attacks, formulates a min-max optimization problem where the inner maximization is to search for the strongest adversarial examples, and the outer minimization reduces the objective caused by those adversarial examples. While PGD training (Madry et al., 2018) and its varieties (Zhang et al., 2019b; Pang et al., 2021; Gowal et al., 2020) have shown promising results, the computation cost of performing adversarial training is relatively high. Faster training methods, such as AdvForFree (Shafahi et al., 2019), YOPO (Zhang et al., 2019a), FastFGSM (Wong et al., 2020), and enhanced FGSM (Chen & Lee, 2020), have been proposed to improve the training efficiency. These faster training methods often generate adversarial examples with weaker attacks, which may lead to lower robust accuracy and even catastrophic overfitting. Besides, Athalye et al. (2018) highlighted the issue of gradient masking in adversarial defenses, which makes it more difficult for attackers to calculate the precise gradient needed to generate effective adversarial examples, resulting in an overestimation of the robustness.

In this paper, we focus on defending against restricted white-box attacks for image classification problems. We revisit the fundamental assumptions of image classification and notice that none of those basic assumptions are satisfied in the real-world scenario. Most pattern recognition systems output only one winning class, which has the highest score among all classes, and as such, the inter-class relationships have been ignored. This implicitly assumes that the ground truth labels are one-hot vectors. However, we identify that one of the factors contributing to the vulnerability of DNNs is the representation of ambiguous examples that contain features from multiple classes using one-hot vectors. Unfortunately, semantic distances among classes are not uniform in real-world data sets, which worsens the robustness of DNNs against unseen examples. Although the optimal label representation for ambiguous examples is unknown, we can generate soft labels using a well-trained model. To achieve this, we present a new algorithm called Low Temperature Distillation (LTD), which enables the target model to learn the inter-class relationships from a prior model. Unlike defensive distillation (Papernot et al., 2016), LTD uses a relatively low temperature in the teacher model and fixed, but different temperatures for the teacher and student models. This modification boosts the model's robustness without encountering the gradient masking problem.

We conduct experiments on CIFAR10; CFAR100 and ImageNet data sets using Wide Residual Network (WRN) family or ResNet-50 to evaluate the effectiveness of the proposed methods. We compare LTD with competitors from the public leaderboard Robustbench (Croce et al., 2021). The experimental results show that LTD using the WRN-34-10 architecture achieves 55.09% robust accuracy without additional data. Combining with Adversarial Weight Perturbation (AWP) (Wu et al., 2020), LTD can achieve 58.19% robust accuracy. For CIFAR100 dataset, LTD achieves 31.13% robust accuracy, which is almost the best result without additional data. For ImageNet, LTD obtains 42.08% robust accuracy, which is about 4% improvement with the same network architecture.

The rest of this paper is organized as follows. Section 2 reviews the related works. Section 3 revisits the fundamental assumptions of image classification and gives an explanation about why the soft labels give more robust training results than one-hot vectors. Section 4 presents the proposed algorithm and its implementation. Section 5 shows the experimental results. The conclusion and future work are given in the last section.

## 2   Related Works

In this section, we give an introduction to the related works. Three topics will be covered. First are the adversarial attacks and defenses. Second is the quality of gradient, used to evaluate the robustness of defense strategies. Last is the framework of knowledge distillation.

### 2.1   Adversarial Attack

The original definition of adversarial examples refers to a modified image that is imperceptible to the human eye but can deceive classifiers. To satisfy the constraints, the distance between the modified image and

the original image should be close to each other. Specifically, the classifier correctly classifies the original examples but the corresponding adversarial examples are misclassified. All adversarial examples are in set $\mathcal{S}$, which is formally defined as follows,

$$\mathcal{S} = \left\{ x' \left| \begin{array}{l} \arg\max Z(x;\theta)_i = \arg\max y_i \\ \arg\max Z(x;\theta)_i \neq \arg\max Z(x';\theta)_i \\ ||x' - x||_\infty \leq \epsilon \end{array} \right. \right\}, \tag{1}$$

where $x$ and $x'$ denote the original examples and the corresponding adversarial examples respectively; the victim classifier $Z$ is a function of an input image and model's weights $\theta$, and $\epsilon$ is the allowed distance in $L_\infty$ space. The classifier outputs the logit $q_i$ for class $i$ and predicts the corresponding label $h$ by the function

$$h = \arg\max q_i = \arg\max Z(x;\theta)_i.$$

Adversarial examples are often created using attacks that compute the gradient's ascending direction to increase the objective loss. For instance, FGSM attack (Goodfellow et al., 2014) generates adversarial examples as

$$x^{\text{FGSM}} = \mathcal{P}(x + \alpha\text{sign}(\nabla L(x, y))), \tag{2}$$

where $L$ is the objective loss, $\alpha$ is the step size, and $\mathcal{P}$ is the projector which ensures that $x^{\text{FGSM}}$ remains in the feasible set. Similarly, PGD attack generates adversarial examples by iteratively running $m$ times FGSM with smaller step-size $\alpha/m$ to get stronger adversarial images.

## 2.2 Adversarial Training

The standard adversarial training (Madry et al., 2018) is designed to defend against adversarial examples, which can be formulated as

$$\min_\theta \mathbb{E}_{x \sim \mathcal{D}} \max_{\tilde{x}:D(\tilde{x},x)<\epsilon} L(Z(\tilde{x};\theta), y). \tag{3}$$

where $x$ lie in the given distribution $\mathcal{D}$; $D(\tilde{x}, x)$ is a distance function for $\tilde{x}$ and $x$. The goal of (3) is to minimize the losses caused by the strongest adversarial examples; however, since the strongest adversarial examples are indeterminable in advance, the inner maximization is usually replaced by known attacks in practice.

A follow-up work, called TRADES (Zhang et al., 2019b), suggested that the worst-case loss in (3) cannot be optimized effectively. Instead, the adversarial loss is decomposed into the natural loss and the boundary loss:

$$L_{\text{TRADES}}(x, x', y) = L(Z(x;\theta), y) + \lambda\Delta L(x, x', y; \theta), \tag{4}$$

where $L(Z(x;\theta), y))$ is original objective in (3) and $\Delta L(x, x', y; \theta)$ is usually the KL divergence as a regularization term. The first term in (4) maximizes the model output distribution between natural data and its corresponding label, while the second term encourages the output distribution to be smooth and pushes the decision boundary away from given examples. The major benefit of KLD loss is label-free. Recent works have shown that KLD can be used to improve the robustness with extra unlabeled data (Alayrac et al., 2019; Carmon et al., 2019; Gowal et al., 2020).

Although adversarial training allows the objective used to generate adversarial examples to be different from the objective used to minimize losses, the choice of the objective remains crucial for the effectiveness of the approach. A previous work (Gowal et al., 2020) investigated several combinations of the inner maximization and the outer minimization, and achieved better robust accuracy under their experiment configurations by feeding the adversarial examples generated by cross-entropy to TRADES where the objective to be minimized is Eq (4) while the original TRADES produces adversarial examples with KLD loss.

The original adversarial training computes adversarial perturbation in input space and minimizes the loss caused by those perturbations. However, the vulnerability may come from weights in hidden layers. AWP (Wu et al., 2020) suggested that the descent direction should be composed of gradients from adversarial perturbation in the input space and in the weight space. This approach can be integrated with existing works, and the empirical results showed updating the weights with composed gradients has higher robustness.

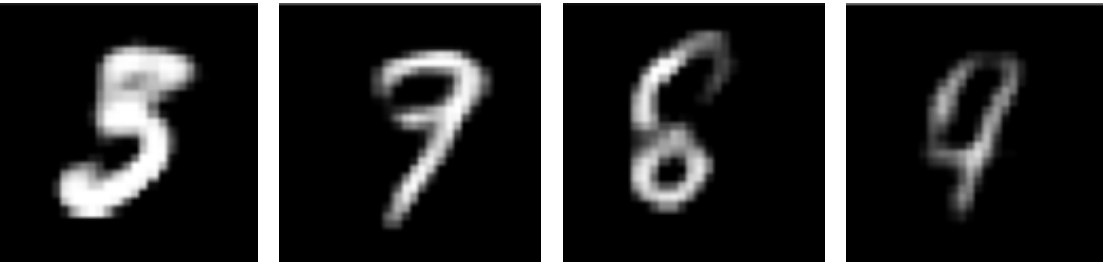

Figure 1: Ambiguous images in MNIST dataset.

Adversarial training has been widely studied and implemented in various forms. However, the derivative of the used objective with respect to the given image is a crucial factor in adversarial training, as it provides the direction for computing the perturbations that are added to the input image. If the gradients are not informative or are masked, adversarial training may fail to generate strong enough adversarial examples to defend against adversarial attacks. Furthermore, it may lead to an overestimation of robust accuracy against gradient-based attacks. This phenomenon is known as gradient masking, which can significantly impact the effectiveness of adversarial training.

To evaluate the quality of the gradients, Athalye et al. (2018) proposed five rules that can be used to determine whether a model has a gradient masking problem. If a model violates any of these properties, it is highly likely to have a gradient masking problem. Similarly, Carlini et al. (2019) provided valuable guidelines for designing a robust adversarial training procedure, summarizing potential pitfalls that should be avoided. These guidelines are essential for ensuring the quality of gradients and avoiding masking problems in adversarial training, leading to more effective and robust models.

### 2.3 Knowledge Distillation

Knowledge distillation (Hinton et al., 2015), originally designed for training smaller models for deployment on edge devices, has now become a foundation for numerous algorithms (Gou et al., 2020). The core concept of knowledge distillation is to provide meaningful label representations from a pre-trained model, known as the teacher model, which enables the target model to learn useful features that may not be shown from ground truth labels. The information distilled from the given model is called soft labels, denoted by $p$, for a given temperature $T = \tau$,

$$p_i^{T=\tau} = \frac{\exp(q/\tau)_i}{\sum_{j=1}^{k} \exp(q/\tau)_j}, \tag{5}$$

where $q$ is the logit computed by the given model. The target model is trained by the following loss function:

$$L = -\sum_{i=1}^{k} y_i \log p_i^{s,T=1} + \lambda \sum_{i=1}^{k} p_i^{t,T=\tau} \log \left( \frac{p_i^{t,T=\tau}}{p_i^{s,T=\tau}} \right), \tag{6}$$

where $y$ is the given one-hot labels; $p^{s,T=\tau}$ and $p^{t,T=\tau}$ are labels given from the target model and the teacher model using the temperature $\tau$ respectively, and $\lambda$ is a scaled factor. The first term in (6) is ordinary categorical loss and the second term is KL divergence (KLD). In Hinton et al. (2015), the authors claimed that the magnitudes of the gradient of KLD loss in temperature $\tau$ is scaled by $1/\tau^2$ and the proper $\lambda$ is $\tau^2$ to balance two losses.

## 3 Data Labeling

For a k-class classification problem, three implicit assumptions are required: closed-world assumption, independent and identically distributed (i.i.d) assumption, and clean and big data assumption (Zhang et al., 2020b). The closed-world assumption supposes that the number of the class $k$ is predefined and all examples must come from the predefined class. For most pattern recognition systems, the models receive an image and

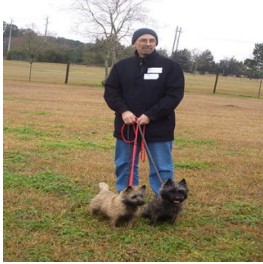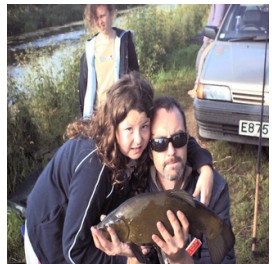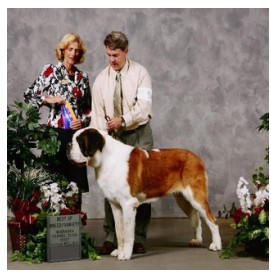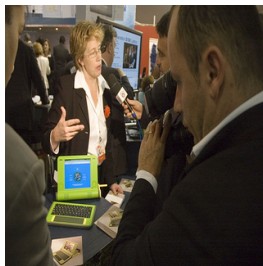

Figure 2: Images with multiple objects in ImageNet dataset.

return a winning class that has the highest score among all classes. The i.i.d assumption supposes that all samples on $\mathcal{D}_{\text{train}}$ or $\mathcal{D}_{\text{test}}$ are drawn from an identical distribution. Under the i.i.d assumption, the objective can be approximated by empirical risk from observed samples on $\mathcal{D}_{\text{train}}$. The clean and big data assumption supposes that all collected data should be well-labeled and large enough for covering the population.

In fact, none of the above assumptions are satisfied in a real-world scenario. Figure 1 shows some images in MNIST dataset are located in the boundary of two classes in semantic space. It is acceptable for those images to be labeled as either of two classes or both classes. Figure 2 shows images with multiple objects in ImageNet dataset. Moreover, semantic distances among classes are not uniform. For example, *automobile* and *truck* are two similar classes in CIFAR10 dataset. For ImageNet dataset, *sunglass* (n04355933) and *sunglasses* (n04356056) are duplicated. *laptop* (n03642806) and *notebook* (n03832673) in ImageNet dataset are identical in semantic space but *notebook* may refer to a book of plain paper. Previous studies argued that the wrong annotation procedure might cause performance degradation (Beyer et al., 2020; Tsipras et al., 2020).

Although the original training procedure using softmax cross entropy loss (SCE) with one-hot vectors achieves good performance with low empirical loss, there are still some issues that need to be addressed. One problem is that the trained model tends to make overconfident predictions on ambiguous samples or poor predictions on samples drawn from different distributions. This phenomenon, known as overfitting, has been observed in both natural and adversarial training (Rice et al., 2020). Furthermore, when minimizing SCE loss with one-hot vectors, the probabilities of the rest of the classes aside from the selected one tend to fade. Consequently, it might lead to a situation where two models have the same accuracy, but one model misclassifies more trivial samples than the other. In this scenario, we would consider the latter model to be superior, despite its high misclassification rate in the ambiguous area. This evidence suggests that one-hot representations are insufficient for delivering information from multiple classes in those ambiguous images.

To break the closed-world assumption, it is essential to have a precise metric for estimating distribution mismatch. Here, we measure the discrepancy gap of a model $Z$ by the closeness between the output probability $p$ and the oracle probability $y^g$ with another widely used function Kullback–Leibler divergence (KLD).

**Definition 1.** *Let $\mathcal{D}$ be the set of all data to be classified, $x$ be an instance in $\mathcal{D}$, $y^g(x)$ be the oracle distribution of $x$, and $p$ be the probability outputs by $Z$ of $x$. The discrepancy of $Z$ is defined as the following expression,*

$$G(Z) = \mathop{\mathbb{E}}_{(x,y^g)\sim\mathcal{D}} \left[ \sum_{i=1}^{k} y_i^g \log \frac{y_i^g}{p_i} \right] = \mathop{\mathbb{E}}_{(x,y^g)\sim\mathcal{D}} \left[ -\sum_{i=1}^{k} y_i^g \log p_i + C \right], \tag{7}$$

where $C$ is a constant. It is worth mentioning that solving the classification problem using SCE loss is a special case of minimizing KLD distance by replacing the oracle distribution with one-hot vectors. The above metric provides valuable insight into the performance of a model $Z$. Specifically, if $G(Z)$ is small, the model is better since its output is closer to the oracle distribution.

As mentioned in Section 3, one-hot labels is not reliable as the labels obtained from the oracle distribution. One approach is to approximate the oracle distribution using knowledge distillation frameworks (Hinton et al., 2015). Even though the training data are still biased, the soft labels, if correctly generated, can reflect

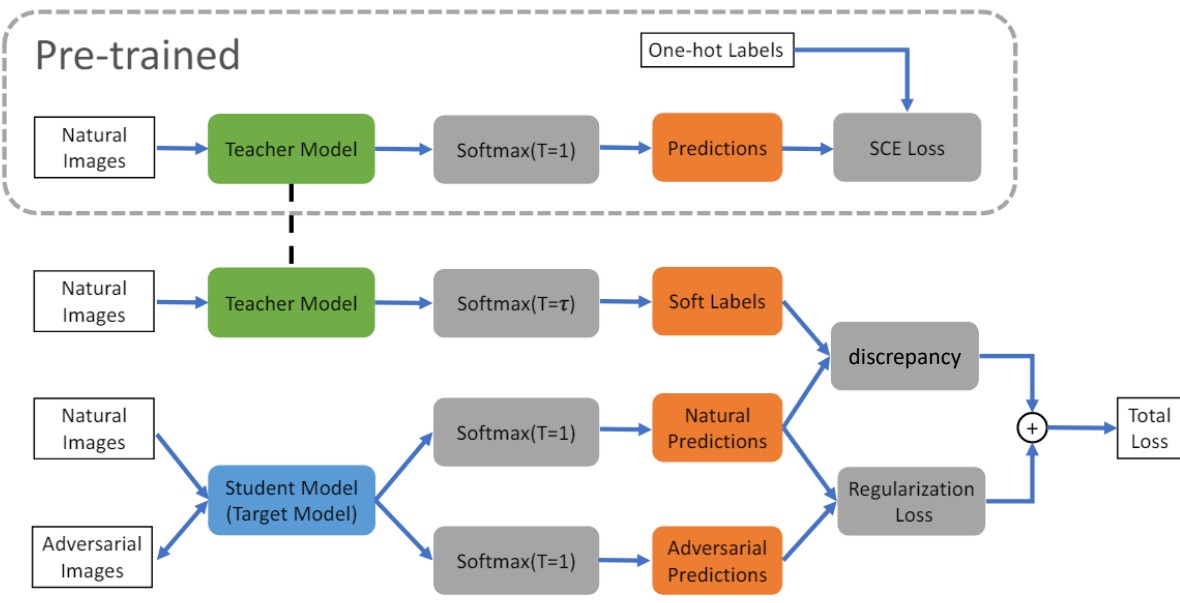

Figure 3: Teacher-target training architecture.

the underlying distributions, including inter-class relationships and precise representation of ambiguous examples, and push the decision boundary toward the optimal one.

The strength of knowledge distillation lies in its flexibility to select the optimal temperature that can be adapted to various types of real-world data sets as shown in Figure 1 and 2. To validate the usefulness of the soft label, we conducted a theoretical analysis for binary classification with a single example, which revealed that the required constraints for soft labels to outperform one-hot vectors are not particularly strict. However, the discrepancy in Eq (7) is an expected value over all examples in the data set, meaning that some examples that violate the constraints are allowable. We further performed simulations on classification problems using synthesized data, which contains only 3% of ambiguity data and there are two models, model 1 and model 2, which are trained using one-hot labels and soft labels respectively. Our analysis and simulations can be found in Appendix B. The analysis shows four important results. First, even a small proportion of noisy data might play a crucial role in degrading the performance of classifiers. Second, the commonly used one-hot label distribution has limitations. Third, the temperature selection is important in the generation of soft-label. Fourth, the model trained with soft labels has a smaller discrepancy although it has a one percent misclassification rate. Furthermore, if the model trained with soft labels can classify all examples correctly and represent inter-class relations precisely, the discrepancy can be further reduced.

## 4 Algorithm and Implementations

### 4.1 Training Framework

Based on the analysis, we propose a training framework, called Low Temperature Distillation (LTD), whose architecture is shown in Figure 3. Followed by the knowledge distillation framework, the teacher model in LTD is obtained from the normal training procedure using SCE loss with natural images $x$ and one-hot encoded labels $y$. The trained teacher model has high natural accuracy but is poor in robustness.

The target model, which is the student model here, is trained by a multiple objective optimization problem with adversarial data and natural data since several studies have shown that adversarial examples are known to be drawn from a distribution that cannot be mimicked by natural data. Its loss function is defined as

$$L_{\text{LTD}} = L(Z(x;\theta), p^t(T = \tau)) + \lambda \Delta L(x', x; \theta), \tag{8}$$

where $p^t(T = \tau)$ is soft labels obtained from the teacher model with the given temperature $\tau$; $L$ is to shape the distribution of natural examples; $\Delta L$ is to minimize the distribution shift between natural data and adversarial examples, and $\lambda$ is to balance two losses. Previous studies (Pang et al., 2021; Wu et al., 2020; Zhang et al., 2019b) have shown $\lambda$ is about 6.0. During the training stage, adversarial examples are crafted using the target model's information and the original label $y$. The labels for $L$ are the soft labels generated by the teacher model, which can be generated in advance or on the fly. Unlike the previous work, LTD uses a relatively low temperature in the teacher model and employs different, but fixed, temperatures for the teacher and student models. This small but crucial modification provides an effective way to enhance the robustness and maintain natural accuracy simultaneously. Moreover, LTD is an easy-to-implement framework that can be integrated into existing models.

## 4.2 Temperature Selection

By the property of (5), if the temperature of the teacher model is high enough, the distribution gradually becomes uniform. As a result, the inter-class relation is destroyed and the discrepancy defined in Eq (7) is dominated by irrelevant classes. This implies that the assumption about the distribution is incorrect, so there is no need to consider the robustness. However, finding the optimal temperature of the teacher model depends on the data set and model architecture. For instance, commonly used models for CIFAR10 data set have highly confident predictions and their output distribution is close to one-hot distribution. Thus, the temperature should be increased. On the other hand, for ImageNet or large-scale datasets, the original distribution may already be suitable for LTD.

We propose a two-step approach for searching for the optimal temperature for LTD training. In the first step, substitute models are trained using only natural data and soft labels generated from a well-trained model with various temperatures. The natural accuracy of each substitute model is evaluated and compared against a predefined threshold. If a substitute model's natural accuracy is lower than the threshold, the corresponding temperature used to generate its soft labels is set as the upper bound for the temperature range to be considered in the second step. In the second step, the optimal temperature is determined by searching for the temperature that minimizes our proposed loss function, defined in equation (8), using any hyper-parameter optimization algorithm in the temperature range $[1, \tau_{\max}]$. It is important to note that the feasible temperature must be lower than $\tau_{\max}$ since adversarial examples hurt the natural accuracy. By searching for the optimal temperature in this two-step approach, we can ensure that the target model is trained with a temperature that is appropriate for the given data set and model architecture, resulting in improved robustness and natural accuracy.

## 4.3 Comparison with Existing Works

In this section, we briefly discuss the difference among existing works, including the objective to be minimized, the usage of the knowledge distillation framework, and label representations.

**Formulation of Loss Function** The objective stated in Eq (8) is similar to previous works, such as TRADES (Zhang et al., 2019b), AWP (Wu et al., 2020), and Pang et al. (2021). However, we emphasize that there is a fundamental difference between our approach and theirs, which lies in the distribution assumption for natural data. In the previous works, the first term in their formulation uses one-hot labels, which might create bias in the output distribution learned from $L(Z(x; \theta), y)$. This bias may cause the adversarial distribution to converge to an incorrect distribution. Our previous analysis suggests that it might worsen the discrepancy gap between the oracle distribution. Additionally, recent studies have indicated that robust overfitting is a common phenomenon in these works Rice et al. (2020). To mitigate this issue, one useful strategy is early stopping. For example, TRADES terminates the training procedure after the second time of decaying the learning rate immediately in default configurations. In contrast, we mimic the oracle distribution by a naturally trained model with low temperature distillation.

**Defensive Distillation** Defensive distillation (Papernot et al., 2016) applied knowledge distillation for adversarial training. However, its goal still wants to fit the one-hot labels. Instead, the purpose of our training method is to learn the soft labels from the teacher model. Moreover,Previous work has been shown that the principle of defensive distillation suffers from the gradient masking problem (Athalye et al., 2018),

Table 1: Competitors from Robustbench on CIFAR10 (Croce et al., 2021)

| # | paper | architecture | $\text{acc}_{\text{nat}}[\%]$ | $\text{acc}_{\text{AA}}[\%]$ |
|---|---|---|---|---|
| * | Wu et al. (2020) + LTD | WRN-34-20 | 86.28 | 58.19 |
| 1 | Addepalli et al. (2021) | WRN-34-10 | 85.32 | 58.04 |
| 2 | Gowal et al. (2020) | WRN-70-16 | 85.29 | 57.20 |
| * | AWP + LTD | WRN-34-10 | 85.21 | 56.90 |
| 3 | Gowal et al. (2020) | WRN-34-20 | 85.64 | 56.86 |
| 4 | AWP (Wu et al., 2020) | WRN-34-10 | 85.36 | 56.17 |
| * | TRADES + LTD | WRN-34-10 | 85.63 | 55.09 |
| 5 | Pang et al. (2021) | WRN-34-20 | 86.43 | 54.39 |
| 6 | Pang et al. (2021) | WRN-34-10 | 85.49 | 53.94 |
| 7 | Pang et al. (2020) | WRN-34-20 | 85.14 | 53.74 |
| 8 | Cui et al. (2021) | WRN-34-20 | 88.70 | 53.57 |
| 9 | Zhang et al. (2020a) | WRB-34-10 | 84.52 | 53.51 |
| - | TRADES (Zhang et al., 2019b) | WRN-34-10 | 84.92 | 53.08 |

as demonstrated by the CW attack (Carlini & Wagner, 2017). There are two major factors that cause this problem. First, the gradient of images can be formulated as a function of the output probability:

$$\nabla_x L_{\text{SCE}} = (p_t - 1)\nabla_x q_t + \sum_{i \neq t} p_i \nabla_x q_i,$$

where $p_i$ is the probability of class $i$ and $q_i$ is the logit of class $i$. This equation shows the gradient almost vanishes when $p_t$ is close to 1, which makes it difficult for attackers to generate adversarial examples using the gradient in inference time.

Second, the temperatures of the target mode in the training stage ($T_t$) and in the inference stage ($T_i$) are different, where $T_t$ is high and $T_i = 1$. It means that the output probability at high temperature ($T_t$) is in the one-hot distribution, and the magnitude of logits in the inference time is $(T_t/T_i) = T_t$ times larger than that of in training time. Consequently, the largest logit dominates others and the output probability converges to a one-hot vector and falls into the area of gradient-vanishing.

The combination of those two factors makes the gradient masking problem more serious. Therefore, attackers cannot generate adversarial examples by the gradient in inference time. The solution we proposed in this work is using different, but fixed, temperatures for the teacher and student models. Specifically, we only adjust the temperature for crafting soft labels but keep the temperature for the student model fixed. This approach ensures that the magnitude of logits is unchanged and gradient masking does not occur.

**Label Representation** Label smoothing (Müller et al., 2019) and learning from the noisy labels are two commonly used approaches to adjust label representations. The former method, label smoothing, redistributes partial probability from the ground truth class to the rest of the classes to prevent the model from overfitting to the one-hot distribution. In contrast, the latter method, learning from noisy labels, applies a stochastic process on labels without considering the context in images, aiming to simulate noises in real-world scenarios. While both methods can achieve the same goal, generating soft labels from a well-trained model can preserve the inter-class relationship. A better label representation is to replace the one-hot label with the distribution which can reveal the probability of the classes related to the target class.

## 5 Experiments

This section presents robustness of LTD against white-box attacks, comparison with with other methods on CIFAR10; CIFAR100, and ImageNet datasets, and ablation studies in the temperature and $\lambda$. The full experiment configurations are presented in Appendix A.

Table 2: Competitors from Robustbench on CIFAR100 (Croce et al., 2021)

| # | paper | architecture | $\text{acc}_\text{nat}[\%]$ | $\text{acc}_\text{AA}[\%]$ |
|---|---|---|---|---|
| * | Wu et al. (2020) + LTD | WRN-34-10 | 64.32 | 31.13 |
| 1 | Cui et al. (2021) | WRN-34-20 | 62.55 | 30.20 |
| 2 | Gowal et al. (2020) | WRN-70-16 | 60.86 | 30.03 |
| 3 | Wu et al. (2020) | WRN-34-10 | 60.38 | 28.86 |

Table 3: Competitors from Robustbench on ImageNet (Croce et al., 2021)

| # | paper | architecture | $\text{acc}_\text{nat}[\%]$ | $\text{acc}_\text{AA}[\%]$ |
|---|---|---|---|---|
| * | LTD | WRN-50-2 | 68.10 | 42.08 |
| 1 | Salman et al. (2020) | WRN-50-2 | 68.46 | 38.14 |
| * | LTD | ResNet-50 | 62.40 | 36.82 |
| 2 | Salman et al. (2020) | ResNet-50 | 64.02 | 34.96 |
| 3 | Engstrom et al. (2019) | ResNet-50 | 62.56 | 29.22 |
| 4 | Wong et al. (2020) | ResNet-50 | 55.62 | 26.24 |
| 5 | Salman et al. (2020) | ResNet-18 | 52.92 | 25.32 |

## 5.1 White-box Robustness

We evaluate the robustness by AutoAttack (AA) (Croce & Hein, 2020), includes APGD-CE, APGD-T, FAB-T, and Square attack. APGD-CE, APGD-T, and FAB-T are PGD-based attacks with different objectives or updating rules. Since most defenses generate adversarial examples by a specific attacking algorithm, the target model may overfit the given attacking algorithm. Consequently, the target model may be defeated by others algorithms and the robust accuracy has a significant dropping eventually. In Athalye et al. (2018), the authors claimed if the black-box attacks are more efficient than the white-box attacks, the defensive methods have gradient masking problems. The square attack is a query-efficient black-box attack that can detect the gradient masking effect. To speed up computing efficiency, AA filters out misclassified data by current attack quickly and the rest of the candidates will be examined with the following attacks. The final accuracy reported by AA ensures the robustness is not overestimated.

We conducted experiments on CIFAR10, CIFAR100, and ImageNet data sets to evaluate the effectiveness of our proposed approach. Table 1 shows the experimental results of CIFAR10, Table 2 presents the results of CIFAR100, and Table 3 gives the results of ImageNet. In these tables, $\text{acc}_\text{nat}$ represents the accuracy on natural data, and $\text{acc}_\text{AA}$ represents the robust accuracy against AA attack. The orders of the methods are based on their robustness accuracy $\text{acc}_\text{AA}$. The numbers in # denote the original rankings of other methods in RobustBench (Croce et al., 2021), and the items with ∗ denote our results. We also included the results of TRADES at the bottom row for comparison.

As can be seen, for CIFAR10 data set, combining LTD with TRADES increases its robustness from 53.08% to 55.09%. Furthermore, integrating AWP with LTD improves its robust accuracy from 56.17% to 56.90% using WRN-34-10. If we use WRN-34-20 with AWP and LTD, the robustness can achieve 58.19%, which is the best result among all the methods. Combining AWP with LTD also improves the robust accuracy for CIFAR100 data set from 28.86% to 31.13% using WRN-34-10, which is the best result without using extra data, and the model size is smaller than that of other competitors. Similarly, the robustness for ImageNet data set is improved significantly from 38.14% to 42.08%.

The experimental results show that LTD is an effective method for improving the robustness of deep learning-based models, particularly for ImageNet data set, which contains many ambiguous images and images with multiple objects. The key difference between the original models, TRADES or AWP, and LTD lies in the labels used for training. The former use one-hot vectors as labels, while LTD replaces them with soft labels generated from the teacher model with low temperature. The results are consistent with our assumption described in Section 3. The data sets contain a lot of ambiguous images and images with multiple objects. The soft label is a better representation of those examples.

Table 4: Ablation study on temperature

| | $acc_{nat}[\%]$ | $acc_{AA}[\%]$ |
|---|---|---|
| TRADES | 84.92 | 53.08 |
| T=1.0 | 84.51 | 54.38 |
| T=2.0 | 84.96 | 54.90 |
| T=3.0 | 85.48 | 55.03 |
| T=5.0 | **86.20** | **55.09** |
| T=8.0 | 85.23 | 54.63 |
| T=10.0 | 84.72 | 53.56 |
| T=12.0 | 77.45 | 43.85 |
| T=15.0 | 94.72 | 0.00 |
| T=20.0 | 94.39 | 0.00 |
| T=50.0 | 86.63 | * |

(a) Ablation study on CIFAR10

| $\lambda$ | architecture | $acc_{nat}[\%]$ | $acc_{AA}[\%]$ |
|---|---|---|---|
| 3.0 | WRN-34-10 | 87.66 | 56.39 |
| 4.0 | WRN-34-10 | 86.61 | 56.61 |
| 5.0 | WRN-34-10 | 85.81 | 56.72 |
| 6.0 | WRN-34-10 | 85.15 | 56.91 |
| 7.0 | WRN-34-10 | 84.97 | 56.20 |

(b) Ablation study on CIFAR100

| $\lambda$ | architecture | $acc_{nat}[\%]$ | $acc_{AA}[\%]$ |
|---|---|---|---|
| 3.0 | WRN-34-10 | 66.67 | 30.63 |
| 4.0 | WRN-34-10 | 65.16 | 30.83 |
| 5.0 | WRN-34-10 | 63.85 | 31.05 |
| 6.0 | WRN-34-10 | 63.31 | 31.13 |
| 7.0 | WRN-34-10 | 63.06 | 30.94 |

## 5.2 Temperature Selection

We used the model WRN-34-10 and CIFAR10 data set and trained the model using TRADES+LTD with different temperatures in the teacher model to verify the importance of temperature selection. As mentioned in Section 4.2, the best temperature is selected in two steps. As shown in Table 4, we searched for the best temperature in the range of $[1.0, 50.0]$, excluding the upper end since its natural accuracy is 86.63% which is too low to accept. The experimental results presented in Table 4 demonstrate that the best temperature for the teacher model is 5.0. As the temperature increases, the robust accuracy decreases due to the fact that irrelevant classes receive partial probability from the target class, which violates our assumption. This leads to gradient masking, and we can observe that the robust accuracy in the training phase is almost 100%, but it cannot defend against stronger attacks or unseen attacks. AA identifies the occurrence of gradient masking when the temperature is higher than 15.0. This phenomenon also has been observed in defensive distillation, where the magnitude of logit is significantly altered.

It is worth emphasize that our proposed method is superior to TRADES across various temperature selection. This result suggests that the one-hot assumption seems to be overconfident on real-world data sets. There are several distribution assumptions can achieve the same or above performance with the existing criteria. Instead, soft label representations generated by the teacher models with low temperature are good choices although the used temperature is not the optimal. Moreover, the above phenomenon is consist with the simulated results.

## 5.3 Ablation study

This experiment investigates the influence of the choice of various $\lambda$ for natural accuracy and robust accuracy. We conducted the ablation study with LTD+AWP on CIFAR10 and CIFAR100 data sets. Table 5a and Table 5b show the results on CIFAR10 and CIFAR100 data sets respectively. As can be seen, the natural accuracy decreases when the value of $\lambda$ increases while the lowest natural accuracy occurs when $\lambda$ is 7.0 on both data sets. These results suggest the best choice of $\lambda$ must be lower than 7.0. Meanwhile, the best robust accuracy occurs when $\lambda$ is 6.0 which is consistent with previous works Wu et al. (2020); Gowal et al. (2020). However, if we prefer higher natural accuracy, the optimal configuration of $\lambda$ can be a smaller value.

Ideally, $\lambda$ should be instance-dependent and its value can be increased to enhance the robustness of some well-classified examples. However, $\lambda$ was predefined in those related works and the optimal value is identical. A possible reason is that those experiments adopted a similar training policy and the same network architecture.

### 5.4 Discussion

The proposed training framework, which replaces the one-hot labels with soft labels, achieves better robustness than the original training framework. We emphasize that the proposed method is the first method that effectively mitigates the gradient masking issue that has been observed in defensive distillation. This approach is suitable for the classification problem which violates closed-world, or clean and big data assumptions. An excellent example of such a problem is ImageNet, where our approach demonstrated a significant improvement of about four percent in terms of robustness.

**Influence of Architecture** Currently, the most popular models used in certified submissions on Robust-Bench are ResNet, WRN series and vision transformer (ViT). ResNet-18 is a standard baseline for evaluating the robustness of lightweight models. WRN series are widely used models to make a fair comparison, which ensures that improvement is gained from the proposed models. The robustness of ViT is controversial since ViT-based generally requires more training examples. It is unclear whether the robustness takes advantage of big data or the special blocks in the network. A recent work introduced a specific adversarial attack for self-attention mechanism Joshi et al. (2021). Although the influence of architecture on the proposed method is unknown, this work followed the training procedure proposed by TRADES and AWP, and achieved better robustness with the same or smaller model size. Additionally, the robustness on ImageNet data set is a challenging task that has not been widely studied yet, such as the initial learning rate, batch size, weight decay, or other hyper-parameter configurations for adversarial training. We believe that the empirical results demonstrated the effectiveness of the proposed methods on various data sets.

**Knowledge Distillation** In this paper, we revisited knowledge distillation frameworks for adversarial training and provided a practical solution to correct the gradient masking issue. These results bring a chance that a robust but lightweight or compressed model can be obtained by using knowledge distillation frameworks. However, the robustness evaluation is more sophisticated for compressed models, since the target models may be defeated by adversarial examples crafted by unpruned models.

We acknowledge that the selection of the optimal temperature for knowledge distillation depends on the complexity of the target dataset. If the examples can be classified accurately without ambiguity, there is no need to modify their label representations. However, for ambiguous examples, the temperature should be dynamically adjusted based on their confidence. In our work, we applied a global temperature for all examples, which can be refined in future works. A potential solution to this issue is to use an auxiliary classifier to identify ambiguous examples and use the output information as a metric for designing temperature adjustment strategies.

Additionally, the choice of the teacher model is an open problem. We chose to train the teacher model naturally, rather than adversarially, because adversarial training may not necessarily produce good teachers in our framework. Adversarial training tends to prioritize robustness over the accuracy, and as a result, the natural accuracy of the teacher model may be relatively low. We experimented with using an adversarially trained model as the teacher, but found that the trained student models achieved lower natural accuracy. For the CIFAR10 dataset, the natural accuracy of the student model was lower than 82%, which is generally unacceptable. While adversarially trained models may extract underlying features that can improve robustness, several issues need to be overcome before they can be used effectively as teacher models in our framework.

**Label Representation** Estimating the oracle distribution is a crucial problem, but LTD is not the only solution to predict the oracle distribution. Several previous works have proposed alternative solutions. AVmixup (Lee et al., 2020) augmented training examples by defining virtual examples, utilizing linear interpolation for both input examples and their labels. While the model trained by AVmixup can mitigate the overfitting issue, it cannot defend against CW attacks or PGD attacks with different objectives, especially for CIFAR100 and more sophisticated data sets. On the other hand, CCAT (Stutz et al., 2020) further suggested that the label representations should be calibrated by the strength of adversarial examples. Another approach

suggested by Chen et al. (2021) is to train the model from two teachers, one being a naturally trained model and the other being an adversarially trained model, which can alleviate overfitting. Ensemble training may also be a proper solution to avoid the overconfidence problem, but there is a lack of systematic studies for designing teacher models, including the total number of teacher models, the training policy, and the used architectures for each teacher model.

**Adversarial Training with Additional Data** The most effective strategy for improving the robustness on CIFAR10 or CIFAR100 data sets is using additional examples from external data sets. The major reason is that traditional supervised learning almost ignores low-frequency data so DNNs cannot recognize them correctly. Adding extra data can cover the data in the regime of low frequency in the original training set. UAT (Alayrac et al., 2019) showed TRADES on CIFAR10 with 200,000 additional images improved the robustness significantly. In UAT, the authors also concluded that selecting a subset from the additional images properly has better robustness than that using entire additional images. The follow-up work RST (Carmon et al., 2019) designed a special classifier to select relevant images, which joined into the adversarial training set, from another dataset.

However, this type of approach is excluded from our approach. We believe that selecting data under the same distribution is an opening problem and the distribution of the joint training examples is shifted entirely since the joint data are selected by an auxiliary classifier in advance. Furthermore, another criticism is that the number of additional images is much larger than that of CIFAR10's training set (50,000 images), and processing those data requires more computational cost. Those similar works, which rely on huge additional images, cannot be extended to ImageNet or other large-scale data sets. In contrast, LTD can achieve significant improvement in the robustness on ImageNet data set. However, we believe that one direction of future work is to identify outliers in advance and exclude those examples from the training set to accelerate training speed.

## 6    Conclusion

The use of one-hot labels as a representation for classification problems was common practice. In this paper, we have demonstrated that one-hot labels are imprecise and one of the vulnerabilities of DNNs arises from ambiguous examples. We have also explored the advantages of soft labels, which can significantly decrease discrepancies between the oracle probability and predictions from a model trained by soft labels. To train robust DNNs, we proposed a modified knowledge distillation framework that utilizes soft labels with a properly low temperature. By using different temperatures for the teacher and student models, we avoided the issue of gradient masking, while the soft labels were able to represent smoother probability distributions among classes. The experimental results on CIFAR10 and CIFAR100 data sets show that our approach, when combined with AWP, achieves robust accuracy of 58.19% and 31.13% respectively. For ImageNet data set, our approach gains about 4% improvement on the robustness. These results provide insights into how to label annotations affect the robustness of deep neural networks. We believe that designing a better label representation for real-world scenarios remains an unexplored issue.

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

## A    Experiment Configurations

We implement our algorithm using CUDA 11.3, cuDNN 8.4 and PyTorch 1.11.0 framework with mixed precision.

### A.1    Dataset

CIFAR10 and CIFAR100 contain 50,000 training images and 10,000 testing images. The values of all images value are normalized in range in $[0, 1]$. Each image in training set are cropped randomly with padding 4 pixels on each border and applied random horizontal flip during training procedure.

ImageNet consists of about 1,300,00 training images collected from real world and each image's dimensions are vary. We apply random resized crop with 224x224 during the training phase and center crop with 224x224 in inference phase. Since we assume that adversarial examples and natural images may have different statistical information, the commonly used normalization by subtracting the mean and dividing by standard deviation may not be suitable for this case. Instead, each pixel is normalized in range in $[0, 1]$.

### A.2    Implementations

Network architecture is Wide Residual Net (WRN) (Zagoruyko & Komodakis, 2016) with width 10 and depth 34 for CIFAR10 and CIFAR100 and the network architecture is ResNet-50 for ImageNet. Both teacher model and target model use the same architecture and configurations. The teacher models are trained from scratch using normal training with one-hot labels, and image transformation is used as mentioned in A.1. The natural accuracy of teacher's models are 94.5% or above and 77% or above for CIFAR10 and CIFAR100 respectively.

In CFIAR10 and CIFAR100 experiments, the target models are trained for 120 epochs. The initial learning rate is 0.1 and divided by 10 at the end of 80-th and 100th epoch. The optimizer is SGD with a momentum of 0.9; weight decay to 0.0005 and Nesterov is enabled. Other configurations follow default settings. The adversarial examples are generated by PGD-8 for TRADES (Zhang et al., 2019b) and AWP (Wu et al., 2020).

In ImageNet experiment, the initial learning rate is 0.1 and divided by 10 at the end of 50-th and 90th epoch. The optimizer is SGD with a momentum of 0.9; weight decay to 0.0001; Nesterov is enabled and batch size is 320. The training is completed at 90-th epochs. The adversarial examples are generated by PGD-6.

### A.3    Metric

We measure the robust accuracy under the white-box environment against the AutoAttack (AA) (Croce & Hein, 2020) with default configuration. For CIFAR10 and CIFAR100, the baseline is $\epsilon = 8/255$ in $L_\infty$ norm. For ImageNet, Robustbench (Croce et al., 2021) evaluates the robustness using fixed 5,000 images from validation set within $\epsilon = 4/255$ in $L_\infty$ norm.

## B    Analysis for Temperature Selection in Knowledge Distillation Frameworks

### B.1    Analysis for the Ideal Case

To illustrate the influence of the temperature selection in the knowledge distillation framework and simplify the discussion, we make the following assumptions.

**Assumption 1.** *Let $M$ be a model trained by the discrepancy in (7) using soft labels $y^p$. When the training finishes, the probability $p$ of each input image $x$ output by $M$ equals the corresponding soft label $y^p$.*

This is a reasonable assumption because if $M$ is well-trained, its discrepancy should be less than the given threshold. Eventually, the probability of each image output by the trained model can approximate the given soft label. To simplify the discussion, we just assume the threshold is negligible.

Now we consider a two-class classification problem. For a given image $x$, its oracle soft label is $(y_1^p, y_2^p)$. Without loss generality, we assume that $y_1^p > y_2^p \geq \xi$, where $\xi$ is a small positive number. Furthermore, let $(p_1^{T=t}, p_2^{T=t})$ be the probability of $x$ normalized by softmax. Since the model is making the correct classification, $p_1^{T=t} > p_2^{T=t}$ for any temperature $t$. There are two models, Model 1 and Model 2, which have the same network architecture but are trained by the soft labels that are generated from the teacher model with different temperatures $t_1$ and $t_2$ where $t_2 > t_1 \geq 1$. Using the definition in (7), we can measure their difference of discrepancy:

$$\Delta L^{t_1 \to t_2} = \mathbb{E}_{x \sim \mathcal{D}} \sum_{i=1}^{k} -y_i^p \log p_i^{T=t_2} - \mathbb{E}_{x \sim \mathcal{D}} \sum_{i=1}^{k} -y_i^p \log p_i^{T=t_1} = \mathbb{E}_{x \sim \mathcal{D}} -\sum_{i=1}^{k} y_i^p \log \left( \frac{p_i^{T=t_2}}{p_i^{T=t_1}} \right). \tag{9}$$

We say that the model trained by the soft label using temperature $t_2$ is better than temperature $t_1$ if

$$\Delta L^{t_1 \to t_2} < 0. \tag{10}$$

The following theorem gives a sufficient condition of $\Delta L^{t_1 \to t_2} < 0$.

**Theorem 1.** *With the above assumptions, if for each image, $p_1^{T=t} > p_2^{T=t}$ for any $t$, and*

$$\frac{y_1^p}{y_2^p} \leq \left| \frac{\log(p_2^{T=t_2}/p_2^{T=t_1})}{\log(p_1^{T=t_2}/p_1^{T=t_1})} \right| \tag{11}$$

*then $\Delta L^{t_1 \to t_2} < 0$.*

*Proof.* We start by simplifying $\Delta L^{t_1 \to t_2}$:

$$\Delta L^{t_1 \to t_2} = \mathbb{E}_{x \sim \mathcal{D}} -\sum_{i=1}^{k} y_i^p \frac{p_i^{T=t_2}}{p_i^{T=t_1}} = \mathbb{E}_{x \sim \mathcal{D}} -y_1^p \log \left( \frac{p_1^{T=t_2}}{p_1^{T=t_1}} \right) - y_2^p \log \left( \frac{p_2^{T=t_2}}{p_2^{T=t_1}} \right). \tag{12}$$

Because $p_1$ is decreasing as $T$ increases, and $p_2$ is increasing with $T$, $\log(p_1^{T=t_2}/p_1^{T=t_1})$ is negative and $\log(p_2^{T=t_2}/p_2^{T=t_1})$ is positive. If condition (11) holds, for each image, we have

$$\left| y_1^p \log \left( \frac{p_1^{T=t_2}}{p_1^{T=t_1}} \right) \right| \leq \left| y_2^p \log \left( \frac{p_2^{T=t_2}}{p_2^{T=t_1}} \right) \right|$$
$$-y_1^p \log \left( \frac{p_1^{T=t_2}}{p_1^{T=t_1}} \right) - y_2^p \log \left( \frac{p_2^{T=t_2}}{p_2^{T=t_1}} \right) \leq 0 \tag{13}$$

Aggregating the results in (13) by taking the expectation, we can show that $\Delta L^{t_1 \to t_2} < 0$. $\square$

Generally, the probability distribution of $p^{T=1}$ is close to one-hot, so $p_1^{T=1}$ is close to 1, and $p_2^{T=1}$ is very small. Since the probability is normalized by softmax, as $T$ increases from 1 to $\tau$, $p_1^T$ decreases slowly and $p_2^T$ increases rapidly. Combing with the result of Theorem 1, we know that a low temperature is a good option. For example, we have two soft labels using $t_1 = 1$ and $t_2 = \tau$, for $\tau > 1$. if $p^{T=1} = (0.999, 0.001)$ and $p^{T=\tau} = (0.99, 0.01)$, the right hand side of (11) is about 254. To make (11) hold, we only need $y_2^p > 0.004$, which is not a very strict condition.

We can also apply Theorem 1 to explain that high temperature is not a proper choice. Suppose we have two soft labels from temperature $t_1$ and $t_2$, $1 \gg t_1 < t_2$. When the temperature increases, the distribution becomes uniform gradually, as can be seen from equation (5). Because the magnitudes of all probabilities are close, $p_1^{T=\tau} \sim p_1^{T=1}$ and $p_2^{T=\tau} \sim p_2^{T=1}$, the right hand side of (11) is small. So making the condition hold becomes more difficult. For example, for $t_1 = 5$ and $t_2 = 7$, $p^{T=t_1} = (0.80, 0.20)$ and $p^{T=t_2} = (0.75, 0.25)$, right hand side of (11) is about 3.88. To make the condition hold, we need $y_2^p > 0.23$, which is too large to be possible.

Of course, Theorem 1 is a sufficient condition for $\Delta L^{t_1 \to t_2} < 0$. In reality, if there are few images violating the condition (11), the above arguments about the selection of temperature can still hold because the discrepancy is an expected value over all examples in the data set.

## B.2 Simulation Analysis

The discrepancy in Eq (7) is an expected value over all examples in the data set, meaning that some examples that violate the constraints are allowable. To further demonstrate the superiority of soft labels over one-hot vectors, we conducted experiments using synthetic data in the binary classification problem. We assume that three percent of the data are ambiguous, with oracle probabilities of $(1 - \gamma, \gamma)$, where $0 < \gamma \leq 1$, and the rest of the data are one-hot vectors with probabilities of $(1 - \epsilon, \epsilon)$, where $\epsilon$ is a tiny positive value to avoid the issue of computing cross-entropy with an infinite value. The probabilities output by the first model and the second model are $(1 - \epsilon, \epsilon)$ and $(1 - s, s)$ respectively for all data, where $s$ is a function of temperature. Synthetic results reveal that the discrepancy of the second model is always smaller than that of the first model when $s = \gamma/10$, which simulates soft labels obtained from low temperature. We can simulate the case for high temperatures by increasing the value of $s$. The results are getting worsen when the temperature raising since the distributions gradually become uniform. We obtain similar results when assuming $\gamma$ for each example is a random variable drawn from the interval $[0, 0.1]$.

On the other hand, for a k-class classification problem, we assume the oracle probabilities are drawn from Dirichlet distribution with a given concentration parameter $\alpha_{oracle} = \alpha$. Here, we assume that at least 97 percent of the data whose the largest probability greater than 0.995. The probability obtained from the first model is a one-hot distribution, while the probability obtained from the second model follows another Dirichlet distribution with concentration parameter $\alpha_{soft} = \alpha_{oracle} \times c$, where $c$ is a positive value used to emulate soft labels generated by low temperature. When compared to the model trained with a one-hot distribution, the model trained with soft labels has a smaller discrepancy although it has a one percent misclassification rate. Furthermore, if the model trained with soft labels can classify all examples correctly and represent inter-class relations precisely, the discrepancy can be further reduced.

