# OpenReview forum: "LTD: Low Temperature Distillation for Gradient Masking-free Adversarial Training"
_TMLR — Rejected by TMLR_

### Review · Reviewer_iGw2 · 2023-03-17

**Summary Of Contributions:**

This work proposes Low Temperature Distillation (LTD) to generate soft labels as an alternative to the commonly used one-hot vector labels for image recognition tasks. The authors show that the imprecise representation of ambiguous images with one-hot labels can hinder the learning process and lead to suboptimal solutions. The use of soft labels generated from a well-trained model enhances the robustness of neural network models against adversarial attacks without requiring defensive distillation. Moreover, the authors investigate how to synergize the use of natural data and adversarial examples in LTD. The experimental results demonstrate that the proposed method achieves promising robust accuracy on CIFAR-10, CIFAR-100, and ImageNet datasets without the need for extra unlabeled data. Additionally, the paper addresses the inconsistent parameter problem in batch normalization caused by using both natural data and adversarial examples. Overall, the contributions of the paper include a new label representation method, a modification to the knowledge distillation framework, and experimental results that shed light on the impact of label annotations on robustness in deep neural networks.

**Audience:**

Yes

**Broader Impact Concerns:**

No concerns regarding the broader impact.

**Claims And Evidence:**

No

**Requested Changes:**

1. Clarify the contribution and originality of the proposed method. The authors should clearly explain how their approach differs from previous work in the field and what novel insights their study brings to the table.
2. Reorganize the paper to make the topics more coherent. The authors should provide a clear justification for why they discuss both LTD and the inconsistent problem of the parameters in batch normalization in the same paper. If the topics are indeed related, this relationship should be made explicit.
3. Conduct a more extensive experimental evaluation. The authors should evaluate the proposed method on a wider range of architectures and datasets to demonstrate its generalizability. Additionally, the authors should compare their results to other state-of-the-art methods, including Madry PGD adversarial training and [A] mentioned in the review.
4. Improve the writing quality. The authors should carefully proofread the paper to eliminate grammar mistakes and improve the clarity of the writing.
5. Shorten the paper. The authors should focus on presenting the core message of the paper and avoid unnecessary information that does not contribute to the main contribution.
6. Provide a more in-depth discussion of the limitations and future work. The authors should discuss potential limitations of their approach and suggest future directions for research based on their findings.
7. Address the weaknesses highlighted in the review. The authors should consider the feedback provided in the review and make appropriate adjustments to the paper to address the identified weaknesses.


**Strengths And Weaknesses:**

### Strengths
(+) This work addresses an important issue. The study of adversarial robustness is an important topic and this work studies the impact of label annotations on robustness.
(+) The authors modify the knowledge distillation framework by using a relatively low temperature in the teacher model and different but fixed, temperatures for the teacher and student models to boost the robustness.
(+) The authors provide experimental results that demonstrate the effectiveness of the proposed method on CIFAR-10, CIFAR-100, and ImageNet datasets, achieving more robust accuracy than the compared methods.

### Weaknesses
(-) The originality of this work is limited: In essence, the authors revisit the knowledge/defensive distillation framework and use a relatively low temperature in the teacher model and different, but fixed, temperatures for the teacher and student models.
(-) The authors discuss two seemingly unrelated topics: It is not clear why the authors discuss LTD and the inconsistent problem of the parameters in batch normalization. What do these topics have to do with each other and why should they be treated together in this work?
(-) The experimental evaluation is limited. The authors mainly evaluate their findings on WRN-34-20. This limits the significance of the presented results since the improved performance could be simply a phenomenon only observed in this architecture.
(-) Some comparisons are missing: I am wondering why the authors did not compare to Madry PGD adversarial training? The authors should also consider a comparison with [A], which seems to achieve better performance than this work (see their Table 3).
(-) For the effectively presented content, the paper writing is too long. The authors give lengthy background and present simple dataset samples, which do not contribute to the core message in the paper.
(-) The writing quality is poor: The paper is sprinkled with grammar mistakes.
(-) Lack of discussion on potential limitations and future work: While the conclusion briefly mentions that there is room for improvement in label representation for real-world scenarios, there is no in-depth discussion on potential limitations or future directions for research based on the findings presented.

[A] Adversarial Vertex Mixup: Toward Better Adversarially Robust Generalization; CVPR 2020

---

> ### Author Response · Authors · 2023-03-22
> **response**
>
> We are grateful to you and the reviewers for your valuable comments and feedback on our paper. In this rebuttal, we summarize the main contributions of our work and provide point-by-point responses to your concerns.
>
> Our work proposes a gradient masking-free adversarial training approach based on knowledge distillation framework to solve the limitations of the one-hot label representation in real-world data sets. Compared with defensive distillation, we suggest that employing different fixed, temperatures for the teacher and student models can solve the issue of gradient masking-free which has been addressed by CW attack. We believe that our contribution is significant for future research on adversarial training.
>
> In response to your concerns:
>
> a) We acknowledge that we did not provide a comprehensive comparison with defensive distillation, which may have led readers to think that our novelty is limited. However, our proposed method fully solves the issue of gradient masking for adversarial training with KD, and we will highlight our main contribution and the differences among related works in the revised version.
>
> b) The presentation of inconsistent batch normalization aims to show that the mismatched distribution not only occurs in the label representation but also in image spaces. The models learned from the original distribution and the adversarial distribution, which conditionally manipulates pixel values of images, have two distinct features. However, we will consider removing this section and focus on the label representation to shorten the length of this manuscript and to improve the coherence of the paper.
>
> c) We respectfully disagree our experimental evaluation is limited. RobustBench is a public benchmark whose purpose is to ensure the certified robustness of the proposed methods. As mentioned at the end of Section 2.1 (Adversarial Attack), AutoAttack, the core method adopted by RobustBench, evaluates the robustness and checks the gradient masking issue by combining four attacking strategies. We can do fair comparison among competitive works by following the same procedure and configurations. We have evaluated the performance of our method on CIFAR10, CIFAR100 and ImageNet. We believe that the empirical results that can convince readers that the efficacy of our method.
>
> As shown in RobustBench homepage, almost all submits in RobustBench use ResNet and WRN models. It ensures that the improvements are gained from the proposed training methods instead of the model architecture. Besides, our methods outperform the original PGD adversarial training, which is ranked 67th for CIFAR10 data set on RobustBench, with 44.04% robust accuracy, while our method achieves 56.94% robust accuracy with the same model (WRN-34-10). Therefore, we use two follow-up works, TRADES and AWP, as our baseline on CIFAR10 and CIFAR100 datasets. Adversarial Vertex Mixup [A] is an interesting work, but their results have not been certified by AutoAttack. Instead, the robustness is evaluated by PGD-20, which tends to overestimate the robustness of their method. The robust accuracy against AutoAttck is lower 50% in our evaluation.
>
> d) We agree with your suggestion to shorten the length of our manuscript and correct grammatical errors. We will remove some minor topics and provide more discussion of the limitations and future works.
>
> We hope that our response has addressed your concerns. We hope the manuscript after careful revisions meet your high standards.

---

### Review · Reviewer_XttV · 2023-03-18

**Summary Of Contributions:**

This paper studies the robust overfitting issue in adversarial training. Specifically, the authors found that a one-hot label may hurt the learning process for image recognition and proposed to use the predicted probability (modified by a temperature hyper-parameter) of a teacher model to replace the one-hot label in TRADES loss. The experimental results demonstrate the effectiveness of the proposed method.

**Audience:**

Yes

**Broader Impact Concerns:**

There is no concern about the ethical implications of the proposed method.

**Claims And Evidence:**

Yes

**Requested Changes:**

1. This paper requires significant polishment on writing, especially the redundant preliminaries (weaknesses 1) and incorrect concepts (weaknesses 3).
2. The authors are encouraged to add more experiments (weakness 2) and provide a deeper analysis (weakness 4) of the mechanism of the proposed method.

**Strengths And Weaknesses:**

### Strength
1. The proposed method is simple and easy to understand. The experimental improvements are notable, demonstrating the proposed method's effectiveness.

### Weakness:
1. This paper should be further polished and simplified. There are 12 pages before the proposed method. For example, since the connection between the 0-1 loss, cross-entropy loss, and KL divergence is common sense in the machine-learning community, there is no need to discuss this from Sec 3.1 to Sec 3.2.
2. There are several important related works [R1][R2] that are not listed in this paper. The authors should compare their results with these related works.
3. In my opinion, generalization usually refers to the model's ability to adapt to new, previously unseen data (i.e.,  the performance on test data). It is better to use other concepts/names rather than "generalization" to mean the discrepancy between the probability certainty in Eqn 15.
4. The proposed method has limited technical novelty. In the proposed approach, the authors only replace the one-hot label in the TRADES loss with the predicted probability (modified with a temperature hyper-parameter) from a teacher model. The authors could contribute to this community by analyzing why this method works in theory. Note that the current analysis is heuristic without a theoretical guarantee.

[R1] Tianlong Chen, Zhenyu Zhang, Sijia Liu, Shiyu Chang, Zhangyang Wang. Robust Overfitting may be mitigated by properly learned smoothening. In ICLR, 2021.

[R2] Chengyu Dong, Liyuan Liu, Jingbo Shang. Label Noise in Adversarial Training: A Novel Perspective to Study Robust Overfitting. In NeurIPS, 2022.

---

> ### Author Response · Authors · 2023-03-22
> **response**
>
> Thank you for your comments and feedback on our manuscript. We appreciate your time and effort in reviewing our work. We would like to clarify our main contribution as follows.
>
> Our work focuses on addressing the limitations of the commonly used one-hot label representation in real-world scenarios. We argue that this representation cannot capture the complexity of real-world data. Especially for the classification problems of ambiguous images and images with multiple objects. We propose the use of soft labels, which provide a more smooth and continuous representation for those images.
>
> Knowledge distillation (KD) is a technique for training lightweight models. Defensive distillation is the first work to train a robust model with KD. However, gradient maksing [1] have been addressed that the robustness is overestimated and the this issue arises when using KD in adversarial training scenarios. In contrast, this work indicate that the issue can be corrected by a small modification. Specifically, we show that by employing different, but fixed, temperatures for the teacher and student models when training the student model, can resolve gradient masking issue and achieve better performance on the robustness. This is a significant contribution to the field of adversarial training.
>
> We appreciate your valuable comments, and we are glad to see that you found our work interesting. We would like to address your concerns and provide responses to your suggestions.
>
> a) We agree with your suggestion to shorten the length of our manuscript. We will remove some basic concepts that are common concepts in the deep learning field and focus on highlighting the differences between our approach and defensive distillation. We will also improve the readability of the manuscript by reorganize the structure of content more clearly and concisely.
>
> b) We appreciate your suggestion to discuss the works related to robust overfitting. The main purpose of [R2] is to explain the double descent phenomenon that occurs when the total training epoch is extremely large, exceeding 1000 epochs. However, most adversarial training frameworks terminate the training procedure at around the 200th epoch. On the other hand, [R1] proposed that the robust overfitting can be mitigated by applying label smooth with two teachers. In contrast, our work suggests that the robustness can be improved by utilizing soft labels from a naturally trained model. In the revised version of the manuscript, we will provide a more comprehensive review of the literature about this topic if necessary.
>
> c) We acknowledge your suggestion to rename the term “generalization” in this manuscript. We will rename the terminology “distribution gap” in the revised manuscript to align with the definition in [2].
>
> d) We will provide a clear comparison with defensive distillation in the revised manuscript, highlighting the specific problem that we have addressed with the modified teach-student framework. We believe that our work provides an easy-to-implement and solution to solve the issue of gradient masking in adversarial training with KD.
>
> e) We respectfully disagree our novelty is limited technical novelty. The true that our modification fully solves the gradient masking issue in adversarial training with KD. Our method achieves significant improvements for CIFAR10, CIFAR100 and ImageNet data sets, which implies the generalizability of our approach. We believe that this achievement is an important milestone for future research on adversarial training.
>
> Practically, the oracle probability is unknown in advance. However, we provide a theoretical proof in Section 3.6 that shows that there is a high probability that soft labels are better than one-hot labels. Since we do not assume knowledge of the oracle distribution, we have to introduce a hyper-parameter which adjusts the temperature such that the soft label can fit to the oracle distribution. However, there may be other methods to estimate the oracle distribution. We will address the limitation and more discussion in the revised version.
>
> We hope that the above clarification adequately conveys the main contributions of our manuscript, and that our response has addressed your concerns. Thank you again for your positive comments and valuable suggestions to improve the quality of our manuscript, and we hope the manuscript after careful revisions meet your high standards.
>
> References:
>
> [1] Athalye, A., Carlini, N., & Wagner, D. (2018, July). Obfuscated gradients give a false sense of security: Circumventing defenses to adversarial examples. In International conference on machine learning (pp. 274-283). PMLR.
>
> [2] Recht, B., Roelofs, R., Schmidt, L., & Shankar, V. (2019, May). Do imagenet classifiers generalize to imagenet?. In International conference on machine learning (pp. 5389-5400). PMLR.

---

### Review · Reviewer_aTG3 · 2023-04-06

**Summary Of Contributions:**

The authors propose a variant of TRADES adversarial training that uses soft labels obtained from a teacher model instead of one-hot labels. The method is shown to improve over TRADES and AT-AWP on CIFAR and ImageNet. The paper also includes discussion of why soft labels are more realistic/meaningful than one-hot labels.

**Audience:**

Yes

**Broader Impact Concerns:**

No broader impact concerns, understanding and improving robustness is generally desirable.

**Claims And Evidence:**

Yes

**Requested Changes:**

See above.

**Strengths And Weaknesses:**

Strengths:
- The premise, that one-hot labels might not represent the task difficulty well, is generlal intruiging and important in my opinion.
- Thorough introduction of background concepts such as adversarial training and distillation.
- Significant improvements over TRADES and AT-AWP on RobustBench, without using additional unlabeled data.
- Thorough discussion of differences to similar related work.

Weaknesses:
- Some comments on writing:
-- In Equation (1), is the second condition necessary? The paper evalates using robust accuracy, which means that if an example is not classified correctly it does not matter whether the attack changes the label - it is counted as an error anyway.
-- For Equation (4), I think it is important to highlight that TRADES also uses a different attack compared to standard AT. This is also not discussed in the proposed method which I feel should be addressed.
-- Is the closed-world assumption really demanding one-hot labels? I think this is confusing. In my opinion, the closed world assumption says that the labels stay the same. For _multiclass_ classification, we generally have the assumption that the true underlying class is unique. But this does not mean that this class can be inferred without uncertainty given the limited information in x.
-- Can the authors restate Assumption 1? The second sentence is unclear to me.
- The introduction/background places a lot of weight on the gradient masking problem. However, there are no experiments or discussion of how the proposed method addresses this. For me this is mainly a discussion point in relation to defensive distillation but not too relevant beyond that.
- In relation to gradient masking, I am also not convinved by the statement that a model is only robust if it is robust against adversarial examples _and_ provides "high-quality" gradients. In the end, I do not care how a method achieves robustness (with gradient masking or not). Gradient masking only becomes a problem in terms of evaluation with white-box attacks, which benchmarks such as AutoAttack and RobustBench overcome to some extent.
- I feel the discussion 3.2 is confusing and I am not sure how much value it adds to the method/paper. First, equations (8) and (9) are simply saying that if a model has lower cross-entropy loss against soft labels than another model, then the corresponding temperature is better. But this misses the point that in the proposed method the soft labels also depend on the temperature because the true soft labels will be unknown. So I am not convinced why we should care about the condition of Thm 1. Second, the discussion is still based on the teacher model being trained against one-hot labels (which I suspect from the authors state that the distribution for T = 1 is close to one-hot). Third, the discussion does not distinguish between ambiguous and non-ambiguous examples. In fact, the authors say that for the non-ambiguous cases, they also used a smooth distribution ("the rest of the data are one-hot vectors with probabilities $(1 - \epsilon, \epsilon)$" which is in itself slightly contradicting). Overall, I understand that the authors want to make a case for using soft labels, but I do not find it convincing and adding value for the reader.
- Regarding related work, I want to highlight [a]. [a], in particular, is an important references where adversarial training is performed against soft - in the extreme case, uniform labels for adversarial examples. This could correspond to the extreme temperature case. However, [a] also adapts evaluation and I would be curious if that could be necessary for the proposed method, as well, depending on the temperature.

[a] https://arxiv.org/pdf/1910.06259.pdf

- Regarding the method, why is the teacher trained naturally and not adversarially? Did the authors try?
- Also, the loss of TRADES is changed, so why is it meaningful to stick with $\lambda = 6$?
- Is the TRADES attack still used. If so, is the attack run against soft-labels as well? In this case, if the labels are "really" soft, how would we determine whether mis-classification occurs.
- Evaluation still happens against one-hot labels. In my opinion this is a general weak point of the paper. If we believe that soft labels are more appropriate for a fraction of the examples, we should assume the same on the test set. This means we have to adapt attacks and metrics accordingly. What do the authors think? Did the authors check qualitatively how predictions for adversarial examples on ambiguous examples change?
- There is also no ablation with respect to the teacher model. I feel this is important. There is no guarantee that the teacher model's soft labels are meaningful in any way, especially as selection is still done against one hot labels.

Conclusion:
I very much like the idea of the paper, addressing ambiguous examples in adversarial training. I also appreciate that the improvements are significant. But I am unfortunately not convinced by the story and writing of the paper and I feel key experiments are missing. Essentially, the method works well, but there is little evidence that the improvements are connected to the main story of the paper (ambiguous examples, not the gradient masking story). Given that TMLR generally requires paper to generate value for readers instead of just having novelty and significant improvements, I am not fully convinced this paper is ready for TMLR yet.

---

> ### Author Response · Authors · 2023-04-10
> **response [1/4]**
>
> Thank you for taking the time to review our manuscript and for providing us with your valuable feedback. We have carefully considered your comments and have revised our responses to address each of your concerns more fluently and professionally.
>
> >Q1: In Equation (1), is the second condition necessary? The paper evaluates using robust accuracy, which means that if an example is not classified correctly it does not matter whether the attack changes the label - it is counted as an error anyway.
>
> A1: Thanks for your kind reminder. It is important to note that the original definition of adversarial examples requires that the classifier correctly classify the original examples (the first constraint), but the corresponding adversarial examples are misclassified (the second constraint). Therefore, the second condition is necessary because it ensures that the adversarial examples are truly deceptive the classifier's detection. We will literary clarify the definition of adversarial examples in the revised version to avoid confusion.
>
> >Q2: For Equation (4), I think it is important to highlight that TRADES also uses a different attack compared to standard AT. This is also not discussed in the proposed method which I feel should be addressed.
>
> A2: We agree that it is important to highlight that TRADES uses a distinct objective compared to standard AT. In our paper, we have already mentioned that standard AT and TRADES use two distinct objectives in Eq (3) and Eq (4). Originally, the inner maximization loss used by the standard AT is cross-entropy, while TRADES uses KLD. In our experiments, we used the same attacking configuration as TRADES and AWP, where the inner maximization loss is KLD. However, as pointed out by [R3a2], the inner maximization loss is replaceable, and cross-entropy appears to be a better choice even when minimizing the objective in Eq (4) proposed by TRADES under their configurations. We will provide a more detailed explanation of the attacking configuration in introduction of adversarial training.
>
> [R3a2] Gowal, S., Qin, C., Uesato, J., Mann, T., & Kohli, P. (2020). Uncovering the limits of adversarial training against norm-bounded adversarial examples. arXiv preprint arXiv:2010.03593.
>
> >Q3: Is the closed-world assumption really demanding one-hot labels? I think this is confusing. In my opinion, the closed world assumption says that the labels stay the same. For multi-class classification, we generally have the assumption that the true underlying class is unique. But this does not mean that this class can be inferred without uncertainty given the limited information in x. -- Can the authors restate Assumption 1? The second sentence is unclear to me.
>
> A3: We appreciate your feedback. According to the definition provided by [R3a3], the closed-world assumption means that the total number of classes k is predefined, and all samples must come from these known classes. On the other hand, most image classification problems receive a single image and return the only one winning class, which implicitly assume that the final ground truth labels are one-hot vectors. However, it does not mean that the model must be trained by one-hot vectors. Instead, our paper mimics the inter-class relationship with soft labels. We agree that combining these two assumptions in one sentence may cause confusion. Therefore, we will provide a more detailed and clear explanation of the closed-world assumption in the revised version, and we will clarify the role of one-hot labels in image classification problems.

---

> ### Author Response · Authors · 2023-04-10
> **response [2/4]**
>
> >Q4: The introduction/background places a lot of weight on the gradient masking problem. However, there are no experiments or discussion of how the proposed method addresses this. For me this is mainly a discussion point in relation to defensive distillation but not too relevant beyond that. In relation to gradient masking, I am also not convinced by the statement that a model is only robust if it is robust against adversarial examples and provides "high-quality" gradients. In the end, I do not care how a method achieves robustness (with gradient masking or not). Gradient masking only becomes a problem in terms of evaluation with white-box attacks, which benchmarks such as AutoAttack and RobustBench overcome to some extent.
>
> A4: We appreciate your feedback and agree that the evaluation of gradient masking is only relevant in the context of white-box attacks. Autoattack is a fair and widely used benchmark to verify whether the proposed model is gradient masking-free, and the details have been mentioned at the beginning of the White-box Robustness (Section 5.1). However, we believe that it is still important to highlight this issue in the introduction/background as it is a common problem in adversarial training and can have significant implications for the robustness of the trained models. Meanwhile, in our paper, we have argued that maintaining inter-class relationships is critical to achieving robustness, and knowledge distillation is one of the ways to accomplish this goal. Compared to defensive distillation, our method not only addresses gradient masking but also retains inter-class relationships using the knowledge distillation framework. We will revise the introduction to make these benefits of our proposed method more explicit.
>
> >Q5: I feel the discussion 3.2 is confusing and I am not sure how much value it adds to the method/paper. First, equations (8) and (9) are simply saying that if a model has lower cross-entropy loss against soft labels than another model, then the corresponding temperature is better. But this misses the point that in the proposed method the soft labels also depend on the temperature because the true soft labels will be unknown. So I am not convinced why we should care about the condition of Thm 1. Second, the discussion is still based on the teacher model being trained against one-hot labels (which I suspect from the authors state that the distribution for T = 1 is close to one-hot). Third, the discussion does not distinguish between ambiguous and non-ambiguous examples. In fact, the authors say that for the non-ambiguous cases, they also used a smooth distribution ("the rest of the data are one-hot vectors with probabilities " which is in itself slightly contradicting). Overall, I understand that the authors want to make a case for using soft labels, but I do not find it convincing and adding value for the reader.
> A5: Your comment on the discussion in Section 3.2 highlights concerns regarding the value and clarity of the content. We appreciate your feedback and would like to clarify several points.
>
> Firstly, our goal in the paper was to demonstrate that one-hot vectors are not suitable for real-world data sets, and we provide evidence to support this claim in Figures 1 and 2. Theorem 1 provides an essential rule for evaluating the suitability of different label representations and suggests that a relatively small temperature adjustment is preferable if the constraint is satisfied. We acknowledge that the oracle distributions are unknown in advance, but our simulation results demonstrate that this is not a strict constraint for real-world data sets. Secondly, we replaced the one-hot probability with 1-eps to avoid the issue of computing cross-entropy with an infinite value. Thirdly, in our simulations, we assumed that the data set consists of examples with one-hot probabilities and ambiguous examples with oracle distributions. Although we may not have correctly guessed the true label representation, our man-made soft labels achieved a lower discrepancy gap between the oracle distributions. This is because the discrepancy is an expected value over all examples in the data set. Our simulation results suggest that soft labels may be a better choice when 3% or more of the data are ambiguous. Additionally, a model that outputs soft labels cannot classify all examples correctly, but it has a lower discrepancy compared to a model that outputs one-hot vectors. These results demonstrate the benefits of using soft labels. We believe that the above evidence strongly supports the usage of soft-labels on real-world data sets. However, we are considering removing the discussion of the ideal case and strengthening our statement by using synthetic data that simulates the oracle distribution in real-world data sets. We hope that our clarification provides further insight into our work.

---

> ### Author Response · Authors · 2023-04-10
> **response [3/4]**
>
> >Q6: Regarding related work, I want to highlight [a]. [a], in particular, is an important references where adversarial training is performed against soft - in the extreme case, uniform labels for adversarial examples. This could correspond to the extreme temperature case. However, [a] also adapts evaluation and I would be curious if that could be necessary for the proposed method, as well, depending on the temperature.
>
> A6: We appreciate your suggestion to consider [a] in our related work. We definitively agree that dynamic temperature adjustment is a potential direction of future works. we should lower the temperature of examples with clearly defined labels, while increasing the temperature for ambiguous examples. In our revised manuscript, we will provide a more comprehensive discussion on temperature selection and its potential impacts on the proposed method.
>
> [a] Stutz, D., Hein, M., & Schiele, B. (2020, November). Confidence-calibrated adversarial training: Generalizing to unseen attacks. In International Conference on Machine Learning (pp. 9155-9166). PMLR.
>
> >Q7: Regarding the method, why is the teacher trained naturally and not adversarially? Did the authors try? There is also no ablation with respect to the teacher model. I feel this is important. There is no guarantee that the teacher model's soft labels are meaningful in any way, especially as selection is still done against one hot labels.
>
> A7: Thank you for raising the question regarding the teacher model. We opted to train the teacher model naturally rather than adversarially because adversarially trained models may not necessarily be good teachers in our framework. Adversarial training tends to prioritize robustness over accuracy, and as a result, the natural accuracy of the teacher model may be relatively low. We did experiment with using an adversarially trained model as the teacher, but found that the student models achieved lower natural accuracy when trained with this teacher. For CIFAR10 data set, the natural accuracy of the student model is lower than 82\%, which is not acceptable generally. We acknowledge that adversarially trained models may extract some underlying features that can improve the robustness, but there are some issues should be overcome. we will include a brief discussion on the limitations of the teacher model in the paper.
>
> >Q8: Also, the loss of TRADES is changed, so why is it meaningful to stick with \lambda=6?
> A8: Thank you for your question regarding the value of lambda in our method. The choice of lambda was based on the results of ablation studies, and we found that a value of 6 was a good choice for several implementations, including AWP and [R3a8]. Ideally, lambda could be instance-dependent and its value can be increased to enhance the robustness for some well-classified examples. While in our experiments, it was predefined. We conducted ablation studies on various values of lambda and found that 6 was still the best value, as shown in Table R3-1 and Table R3-2.
>
> Table R3-1 ablation study on CIFAR10 data set
> | \lambda | acc_{nat} | acc_{adv} |
> |:-------:|:---------:|:---------:|
> |   5.0   |   85.81   |   56.72   |
> |   6.0   |   85.15   |   59.91   |
> |   7.0   |   84.97   |   56.20   |
>
> Table R3-2 ablation study on CIFAR10 data set
> | \lambda | acc_{nat} | acc_{adv} |
> |:-------:|:---------:|:---------:|
> |   5.0   |   63.85   |   31.05   |
> |   6.0   |   63.31   |   31.13   |
> |   7.0   |   63.06   |   30.94   |
>
> [R3a8] Pang, T., Yang, X., Dong, Y., Su, H., & Zhu, J. (2020). Bag of tricks for adversarial training. arXiv preprint arXiv:2010.00467.
>
> >Q9: Is the TRADES attack still used. If so, is the attack run against soft-labels as well? In this case, if the labels are "really" soft, how would we determine whether mis-classification occurs.
> A9: We are not sure what the TRADE attack that you mentioned refers to. However, we would like to point out that AutoAttack, the evaluation method we used, involves 19 objectives to evaluate the robustness. Among these objectives, the first objective minimizes the probability of the ground truth class while the rest of the objectives maximize the probability of the specific targeted class. Since the classification problem assumes that there is the only one winning class, the label representations are one-hot for all attacks.
>
> We understand your concern about misclassification under soft label representation. In our work, we can assume that the top-k likely classes outputted by the model for ambiguous examples are acceptable predictions. However, determining the value of k for each image is a challenge, and currently, we do not have a convincing approach to determine it. We agree that top-k accuracy is an alternative metric, but it is instance-dependent.

---

> ### Author Response · Authors · 2023-04-10
> **response [4/4]**
>
> >Q10: Evaluation still happens against one-hot labels. In my opinion this is a general weak point of the paper. If we believe that soft labels are more appropriate for a fraction of the examples, we should assume the same on the test set. This means we have to adapt attacks and metrics accordingly. What do the authors think? Did the authors check qualitatively how predictions for adversarial examples on ambiguous examples change?
>
> A10: Thank you for your question regarding the evaluation of our proposed method. We agree that using one-hot labels for evaluation is a limitation, and the soft label assumption should be considered for real-world data sets. However, we believe that the adaptation of metrics and attacks should be data-dependent and problem-specific.
>
> For example, in our work, we addressed the issue of duplicated classes in ImageNet dataset, where the correct prediction can be one of the duplicated classes. In the case of ambiguous examples, the most likely class can be considered an acceptable prediction. Therefore, we argue that the definition of adversarial examples should not involve those cases as well.
>
> Moreover, AutoAttack, which we used for evaluation, checks whether adversarial examples can change the labels to one of the nine most likely classes. If we exclude those cases, the robustness of our method increases by about 3%, 5%, and 6% for CIFAR10, CIFAR100, and ImageNet datasets, respectively.
>
> In conclusion, we agree that the soft label assumption should be considered for evaluation, and we will investigate potential metrics and attack adaptations for future works. Nonetheless, we believe that the specific problem and data distribution must be considered to determine the appropriate evaluation strategy.

---

### Author Response · Authors · 2023-03-28
**Summary of revision**

We express our sincere gratitude for your valuable feedback on our manuscript, and we have incorporated the suggestions provided. We have carefully revised the manuscript to address your concerns and suggestions. The following are significant changes:

a. We agree that the manuscript needed to be shortened, and therefore we have made significant changes.  We have removed the presentation of inconsistent batch normalization (Sections 3.4, 3.5, and 5.3.1 in the original manuscript). Additionally, we have combined Sections 3.1 and 3.2 in the original manuscript into Section 3.1 (Classification Problem in Real-world Scenario) and Sections 4.1 and 4.2 in the original manuscript into Section 4.1 (Training Framework). We are confident that these modifications have made the manuscript more concise and comprehensible.

b. We have polished Section 3.2 (Oracle Distribution Estimation) and added simulation evidence to reinforce our claim that soft labels are superior to one-hot vectors.

c. We have added a section titled "Comparison with Existing Works" in Section 4.3 to address your concern. We believe that this addition has significantly improved the manuscript's comprehensiveness and contribution to the field.

d. We agree that the original manuscript lacked a depth discussion on potential limitations and future directions. Therefore, we have added Section 5.3 (Discussion) to address these issues. We believe that this addition has improved the manuscript's quality and its potential to contribute to future research.

f. We have taken great care to eliminate any grammatical errors in the manuscript.

Overall, we appreciate your feedback and believe that the revisions we have made have significantly improved the quality of the manuscript. We hope that you find our revised manuscript satisfactory for publication.

Thank you for your time and consideration.

---

### Author Response · Authors · 2023-04-19
**Summary of the revision**

In the revised version of our manuscript, we have implemented several modifications to address the concerns raised by the reviewers.

a. Revised the statement of the closed-world assumption and the definition of adversarial examples for clarity.

b. Reorganized the introduction for adversarial training to establish a stronger connection to the gradient mask issue.

c. Moved the analysis and simulations to Appendix B and provided additional details to our simulations to support the benefits of using soft label representation in real-world scenarios. Also included the results of the temperature selection analysis to reinforce our findings.

d. Added an ablation study in λ to provide further insight into the effects of this parameter on our model's performance.

e. Added more details to the discussions section to provide a comprehensive understanding of our research and its implications.

---

### Decision · Action_Editors · 2023-06-09

**Recommendation:** Reject

**Comment:**

This submission proposes to train an adversarially robust model through a combination of distillation from a conventionally trained teacher and a penalty on the KL divergence between predictions on clean and adversarially perturbed examples. The proposed method outperforms previous approaches that do not make use of distillation.

In their initial reviews, reviewers raised concerns regarding the organization of the paper. In response, the authors performed a major revision, removing an analysis of the impact of batch normalization on adversarial training that the reviewers suggested was a distraction from the paper’s main message and shortening the presentation to make the paper more manageable. In their final assessments, two reviewers lean toward rejection while one leads toward acceptance. The positive reviewer notes that they are “still not entirely convinced by the story.” One of the negative reviewers states that they believe the evaluation is insufficient and additional model architectures are needed to support the claims, whereas the other notes that, despite the paper’s emphasis on the claim that the proposed methodology solves the problem of gradient masking, there is no evaluation to demonstrate this.

Although the efficacy of the technique is reasonably well-demonstrated, the paper heavily emphasizes some claims where evidence is weak (see "Claims"). This can likely be done with new experiments to support these claims or by adjusting the framing to more accurately convey the level of evidence for them. Perhaps a bigger issue is that there is some very relevant literature that combines adversarial training with distillation that is not compared with or even cited, e.g. [1-3], all of which describe methods that differ from the proposed technique, but in subtle ways (see Table 1 in [3]). Although methodological novelty isn't a consideration at TMLR, without experimental comparison against past work, it will be difficult for a reader to know whether the proposed method performs better, worse, or about the same, which diminishes the amount of interest it will hold. Because addressing these concerns would likely require more than a minor revision, I recommend that the authors revise and resubmit this work.

[1] Goldblum, M., Fowl, L., Feizi, S., & Goldstein, T. (2020, April). Adversarially robust distillation. In Proceedings of the AAAI Conference on Artificial Intelligence (Vol. 34, No. 04, pp. 3996-4003).\
[2] Zi, B., Zhao, S., Ma, X., & Jiang, Y. G. (2021). Revisiting adversarial robustness distillation: Robust soft labels make student better. In Proceedings of the IEEE/CVF International Conference on Computer Vision (pp. 16443-16452).\
[3] Zhu, J., Yao, J., Han, B., Zhang, J., Liu, T., Niu, G., ... & Yang, H. (2022). Reliable adversarial distillation with unreliable teachers. ICLR

**Audience:**

Although the proposed technique is potentially of interest, the paper does not compare against closely related previous work, which may make the paper less interesting for individuals who are familiar with this previous work.

**Claims And Evidence:**

Overall, the efficacy of the method in improving adversarial robustness is well-supported. However, the paper makes a few claims that are substantiated only indirectly, and where the chain of reasoning is not quite strong enough.

First, the paper claims that the proposed technique avoids gradient masking, but the only evidence provided is indirect, via the model's robustness to AutoAttack, a concern raised by two of the reviewers. I feel that the details here suggest that there is no gradient masking—low-temperature distillation probably doesn't cause gradient masking, and adversarial training is conducted with PGD, which is unlikely to be effective in conferring robustness to AutoAttack if gradients are masked. However, the claim that the method is "gradient masking-free" is core to the current framing, and should be tested more explicitly.

Second, the introduction states that "one of the factors contributing to the vulnerability of DNNs is the representation of ambiguous examples that contain features from multiple classes using one-hot vectors," but this claim is not fully substantiated—although the proposed technique improves accuracy under adversarial attacks, it's also well-known that distillation enhances clean accuracy.

**Resubmission Of Major Revision:**

The authors may consider submitting a major revision at a later time.